# Dietary Modulation of the Immune System

**DOI:** 10.3390/nu16244363

**Published:** 2024-12-18

**Authors:** Luis Fernando Méndez López, José Luis González Llerena, Jesús Alberto Vázquez Rodríguez, Alpha Berenice Medellín Guerrero, Blanca Edelia González Martínez, Elizabeth Solís Pérez, Manuel López-Cabanillas Lomelí

**Affiliations:** Universidad Autónoma de Nuevo León, Facultad de Salud Pública y Nutrición, Centro de Investigación en Nutrición y Salud Pública, Monterrey 64460, México; luis.mendezlop@uanl.edu.mx (L.F.M.L.);

**Keywords:** immunity, nutrients, inflammation, infection, chronic diseases

## Abstract

Recent insights into the influence of nutrition on immune system components have driven the development of dietary strategies targeting the prevention and management of major metabolic-inflammatory diseases. This review summarizes the bidirectional relationship between nutrition and immunocompetence, beginning with an overview of immune system components and their functions. It examines the effects of nutritional status, dietary patterns, and food bioactives on systemic inflammation, immune cell populations, and lymphoid tissues, as well as their associations with infectious and chronic disease pathogenesis. The mechanisms by which key nutrients influence immune constituents are delineated, focusing on vitamins A, D, E, C, and B, as well as minerals including zinc, iron, and selenium. Also highlighted are the immunomodulatory effects of polyunsaturated fatty acids as well as bioactive phenolic compounds and probiotics, given their expanding relevance. Each section addresses the implications of nutritional and nutraceutical interventions involving these nutrients within the broader context of major infectious, metabolic, and inflammatory diseases. This review further underscores that, while targeted nutrient supplementation can effectively restore immune function to optimal levels, caution is necessary in certain cases, as it may increase morbidity in specific diseases. In other instances, dietary counseling should be integrated to ensure that therapeutic goals are achieved safely and effectively.

## 1. Introduction

Malnutrition endangers the host’s survival by compromising immune response efficacy. This dependence is reinforced by the fact that malnutrition is the leading cause of immunodeficiency globally [1]. The immune system requires significant energy and nutrients, which must be obtained from dietary sources or bodily reserves. Until the 1950s, the relationship between nutrition and infection was primarily attributed to protein deficiency due to its direct impact on antibody production [2]. At that time, the understanding of the immune system was rudimentary and largely based on in vitro studies, which identified plasma cells as the primary producers of humoral immunity. It was not until the 1970s that the association between malnutrition and cell-mediated immunity was established, alongside an acknowledgment of the metabolic consequences of infection, whereby an activated immune response increases energy demands, resulting in elevated basal energy expenditure [3]. By the 1980s, animal studies had demonstrated the relative impact of nutrients on immune system components in controlled environments [4]. In humans, in vitro lymphocyte culture experiments and the habitual consumption of diets deficient in one or more nutrients provided evidence of the specific effects of certain vitamins on immune system components [4]. Key nutrients identified for their significance in the immune response include essential amino acids and fatty acids; folic acid; vitamins A, B6, B12, C, and E; and the minerals zinc, iron, and selenium [1,5]. Later, in the mid-1990s, a biological explanation was offered for the suppression of the immune response associated with diminished energy reserves [6]. Specifically, the atrophy of adipose tissue resulting from malnutrition leads to reduced leptin production, impairing immune function, as this hormone is crucial for the development, proliferation, maturation, and activation of the immune system [7]. Conversely, it was shown that obese individuals produce up to ten times more leptin, resulting in systemic pro-inflammatory effects [8]. During the same decade, it was reported that adipose tissue secretes tumor necrosis factor-alpha (TNF-α) under conditions of obesity. This protein is a pro-inflammatory cytokine that has been causally linked to the development of metabolic syndrome [9]. Subsequently, it was noted that changes in adipose tissue due to obesity not only included cytokine secretion but also the infiltration of macrophages, T cells, and B cells. In the 2000s, it was demonstrated that the macrophages, lymphocytes, and cytokines produced during the inflammatory process in adipose tissue associated with obesity were primarily responsible for the metabolic dysfunction observed in these patients [9]. During that period, the role of chronic inflammation in major degenerative diseases became increasingly recognized, generating significant interest in researching the influence of nutrients and dietary patterns on systemic markers of immune activation, such as C-reactive protein (CRP), interleukin-6 (IL-6), and TNF-α [10]. More recently, studies conducted after 2010 have highlighted the importance of certain nutrients and the gut microbiota in the induction and maintenance of regulatory T cells (Tregs), which are essential for tolerance to self-antigens, dietary components, and commensal organisms, as well as for modulating inflammatory responses [11]. The growing understanding of the interaction between nutrition, gastrointestinal commensals, and the immune system is encompassed within the field of immunonutrition, a multidisciplinary area that is giving rise to strategies for the dietary and nutraceutical management of major diseases related to inflammatory metabolic processes [12].

## 2. Essentials of the Immune System

The immune system is a complex defense network designed to protect the body against harmful agents and is capable of responding to a vast array of antigens. It must constantly distinguish between self and foreign components while tolerating harmless entities from food, the environment, and commensal bacteria [4]. Impairments in immune function are causally linked to the onset, progression, and severity of numerous chronic conditions, including obesity, diabetes, cancer, allergies, irritable bowel syndrome, cardiovascular diseases, and autoimmune disorders [13]. The immune system comprises lymphoid organs and tissues, specialized immune cells, and signaling molecules. Key immunological organs include skin, mucous membranes, and the lymphatic system. The skin and mucosal surfaces act as primary barriers against external antigens and microorganisms. The epithelium forms a biochemical and physical barrier that separates the internal environment from external agents. Mucosal surfaces secrete mucus and other antimicrobial substances to regulate the growth of potential pathogens [14]. For example, lysozyme, found in saliva, the respiratory tract, and tear fluid, breaks down bacterial cell walls. Epithelial cells, through direct and indirect interactions with other innate immune cells, can initiate both local and systemic immune responses. Stomach acid acts as a defense against pathogens in ingested food, while commensal microorganisms on the skin and mucous membranes inhibit pathogen colonization [15]. The bone marrow and thymus are the primary lymphoid organs responsible for producing and maturing leukocytes [5]. In bone marrow, pluripotent hematopoietic stem cells differentiate into either a common lymphoid progenitor, which produces T cells, B cells, and NK cells, or a common myeloid progenitor, giving rise to neutrophils, eosinophils, basophils, dendritic cells, mast cells, macrophages, erythrocytes, and platelet-producing megakaryocytes [14]. With aging, bone marrow is progressively replaced by adipose tissue, diminishing its capacity for immune cell production. Similarly, the thymus undergoes age-related degeneration, as functional tissue is gradually substituted by fat, reducing its ability to mature T lymphocytes [4]. Secondary lymphoid organs include the lymph nodes, spleen, tonsils, and mucosa-associated lymphoid tissue, with the latter being the largest immune organ in the body [15]. When pathogens or other harmful agents bypass physical and chemical barriers, the immune system activates defense mechanisms. The initial response is driven by the innate axis, consisting of macrophages, neutrophils, mast cells, eosinophils, basophils, dendritic cells, and NK cells [16]. These cells act promptly and are typically less specialized, providing broad-spectrum defense. In contrast, the adaptive immune response, while slower to fully activate, often taking days or weeks, is highly specific and has the unique ability to form memory, allowing for faster and more targeted defense upon re-exposure to the same pathogen [16]. Myeloid cells primarily constitute the innate immune system and include monocytes (which are precursors to macrophages), mast cells, dendritic cells, neutrophils, basophils, and eosinophils [14]. These cells exhibit varying degrees of phagocytic capability, allowing them to detect, engulf, and destroy infectious agents via specialized receptors [16]. In contrast, lymphocytes comprise up to 30% of leukocytes and are central to the adaptive immune system due to their ability to produce highly specific cell surface receptors. Upon recognizing an antigen, T and B lymphocytes undergo clonal expansion, generating large numbers of cells that specifically target the pathogen within 5 to 7 days. Many of them persist as memory cells, facilitating rapid and precise responses to future encounters with the same pathogen [16]. T lymphocytes play a crucial role in adaptive cellular immunity, performing three primary functions: assisting other immune cells (helper), regulating immune responses to prevent excessive reactions (regulatory), and directly eliminating infected or damaged cells (cytotoxic). Helper T cells express the CD4 coreceptor, while cytotoxic T cells express CD8, both distinguished by the presence of the T-cell receptor (TCR) [17]. Cytotoxic T lymphocytes specifically target and destroy damaged, infected, or cancerous cells, whereas helper T cells aid B lymphocytes in differentiating into plasma cells, which produce and secrete antibodies. B cells undergo full maturation in the bone marrow, and their primary function is the production of immunoglobulins (Ig), contributing to humoral immunity. [14]. Antigens, which the adaptive immune system specifically recognizes, typically possess a three-dimensional structure that complements immune receptors. For B cells, antibody molecules serve as antigen receptors, and upon their recognition, they differentiate into plasma cells that secrete copies of these molecules, initiating the humoral response [14]. Antigens can consist of proteins, carbohydrates, lipids, nucleic acids, or small chemical groups known as haptens. When antibodies bind to these antigens, they induce processes such as neutralization, lysis, or phagocytosis [15]. NK cells are lymphocytes of the innate immune system due to their use of germline-encoded receptors. NK cells identify and eliminate cells that display abnormal profiles of the major histocompatibility complex (MHC) receptors [18]. Viruses frequently suppress the expression of MHC molecules as a mechanism to evade the adaptive immune response, thus triggering NK cell activation and the subsequent rapid destruction of infected cells. This capability also allows NK cells to recognize and target tumor cells [18]. The immune system secretes soluble molecules, mainly proteins and lipids, that coordinate, activate, and amplify its effector functions. Crucial immune proteins include antibodies, cytokines, chemokines, interferons, the complement system, and acute-phase proteins. Lipid-based mediators comprise eicosanoids, notably prostaglandins and leukotrienes [16]. Plasma cells secrete antibodies of the IgM, IgD, IgG, IgA, and IgE classes, with primary roles in the opsonization and neutralization of antigens. Leukocytes produce various cytokines, such as IL-6, TNF-α, IL-2, transforming growth factor beta (TGF-β), IL-4, IL-5, and IL-10, which contribute to the activation, differentiation, proliferation, or suppression of immune cells [14]. Chemokines, including monocyte chemoattractant protein-1 (MCP-1), macrophage inflammatory protein-1 (MIP-1), and IL-8, direct leukocytes to infection sites or tissue injury [14]. Interferons like IFN-α, IFN-β, and IFN-ε are produced by virally infected cells, acting to reduce protein synthesis and limit replication [19]. The liver synthesizes complement proteins C1–C9, which facilitate the opsonization, attraction, and lysis of pathogen membranes. Additionally, acute-phase proteins—such as CRP, serum amyloid A, plasminogen, fibrinogen, ceruloplasmin, and α1-antitrypsin—serve as opsonins and complement activators [20]. Adipose tissue also contributes to immune function by producing cytokines and adipocytokines, a class of proteins primarily synthesized by adipocytes [9]. This group consists of leptin, adiponectin, resistin, chemerin, visfatin, and apelin, each involved in the activation, differentiation, attraction, or suppression of immune cells (Figure 1) [21]. Prostaglandins (PGE2, PGI2, PGD2, and PGF2α), synthesized in most cells via the enzymatic action of cyclooxygenase (COX) on membrane fatty acids, play roles in inflammation, altering vascular permeability, vasodilation, blood flow, fever, pain, and tissue damage [22]. Likewise, leukotrienes (LTB4, LTC4, LTD4, and LTE4) are produced through the action of the lipoxygenase (LOX) enzyme in leukocytes, influencing vascular permeability, local blood flow, leukocyte chemotaxis, degranulation, and reactive oxygen species production [22]. When these molecules bind to their specific receptors, they generally activate immune cells and components, leading to the release of the transcription factor NF-κB. This protein then translocates to the nucleus and facilitates the expression of over 500 genes, including those encoding cytokines, chemokines, interferons, acute-phase proteins, and related receptors, as well as COX and LOX enzymes [13].

## 3. The Impact of Nutritional Status on Immune System Function

Malnutrition significantly impairs immune function by causing atrophy of bone marrow, resulting in reduced cell proliferation, fewer mature immune cells, and a decreased proportion of lymphocytes. It also leads to thymic atrophy due to diminished thymocyte proliferation and increased apoptosis [23]. Under malnutrition, the thymus produces less thymulin, a hormone essential for T-cell maturation. Similar atrophic effects are observed in lymphoid tissues such as lymph nodes, tonsils, and the spleen, likely due to altered production and maturation of myeloid and lymphoid cells, which contributes to the leukopenia, compromised immunity, and increased infection susceptibility seen in malnourished individuals [24]. Malnutrition also suppresses innate immune defenses, reducing the secretion of lysozyme, gastric acid, cytokines, interferons, antibodies, complement, the microbicidal activities of neutrophils, macrophages, and the levels of plasma cells [5]. Similarly, obesity impairs immune function, despite elevated leukocyte counts and inflammatory mediators. It is associated with increased hospitalizations and complications from infectious diseases. Diet-induced obesity weakens T- and B-cell memory responses, compromising vaccine effectiveness [12]. In obese subjects, immune deficiencies are linked to an increase in fatty tissue within the bone marrow and thymus. Adipocytes in this context experience hypertrophy, hyperplasia, hypoxia, oxidative stress, and cell death, leading to the release of pro-inflammatory cytokines, chemokines, and damage-associated molecules, which subsequently activate NF-κB and sustain a state of chronic inflammation [13]. In obesity, adipocytes produce elevated amounts of leptin, TNF-α, IL-6, visfatin, and resistin, all of which further disrupt metabolic and inflammatory regulation. Most leukocytes have receptors for leptin, explaining its direct impact on immune function [25]. Leptin promotes Th1 polarization, which enhances the secretion of pro-inflammatory cytokines. Conversely, plasma amounts of adiponectin, an anti-inflammatory adipokine, are reduced in obesity. Through its receptor, AdipoR, adiponectin inhibits NF-κB activation and reduces the production of pro-inflammatory cytokines, including TNF-α, MCP-1, and IL-6 [25]. Adiponectin also suppresses the activation of phagocytes, eosinophils, T cells, NK cells, and dendritic cells [21]. Although the precise mechanisms by which obesity compromises immune function remain unclear, it is thought to involve histological changes in lymphoid organs, alterations in adipose tissue, and an imbalance in cytokine profiles [26]. Figure 2 provides a schematic representation of the major histological changes in key lymphoid tissues induced by malnutrition and obesity.

## 4. Diet and Inflammation

Inflammation is a critical component not only in infectious processes, such as acute respiratory syndrome caused by coronavirus (COVID-19), but also in the pathophysiology of several chronic conditions, including obesity, cardiovascular diseases, type 2 diabetes mellitus, autoimmune disorders, Alzheimer’s disease, and cancer [12,13]. Inflammatory markers show a positive correlation with elevated triglycerides, glucose, oxidized LDL, insulin resistance, and hypertension, making them useful in early detection and risk assessment in patients with metabolic diseases. Critical biomarkers of metabolic dysfunction include CRP, leptin, TNF-α, and IL-6 [20]. Research showing that high concentrations of pro-inflammatory cytokines predict disease progression and severity has intensified interest in the dietary modulation of immune activity [10]. In particular, plant-based diets have been associated with reduced incidence of chronic diseases, attributed to their long-term anti-inflammatory effects. Vegans are associated with lower amounts of CRP compared to both vegetarians and omnivorous. Among these two dietary patterns, vegetarians exhibit significantly reduced levels of IL-6, CRP, fibrinogen, and total leukocytes compared to omnivores [27]. The Mediterranean diet is similarly correlated with reductions in plasmatic inflammatory markers, such as IL-6, IL-8, MCP-1, TNF-α, and IFN-γ [28]. Likewise, a traditional Mexican diet shows an inverse relationship with biomarkers of inflammation and metabolic dysfunction [29]. Conversely, adopting a Western-style diet significantly increases circulating concentration of IL-8, IL-6, CRP, and negatively affects metabolic function. Regarding specific food groups, regular consumption of fruits and vegetables generally reduces serum CRP levels [30]. In patients with metabolic syndrome, cruciferous vegetables are linked to lower plasma CRP content [31], while regular berry intake decreases CRP, IL-6, and circulating vascular cell adhesion molecule 1 content, a biomarker of vascular damage [32]. In overweight and obese individuals, whole grains reduce TNF-α levels, and legume intake lowers serum CRP and other metabolic dysfunction markers [29]. The daily intake of monounsaturated and polyunsaturated fat from seeds and nuts reduces endothelial inflammation and the amount of CRP, TNF-α, IFN-γ, IL-1β, IL-6, and stress markers in patients with diabetes or cardiovascular disease. Similarly, olive oil, recognized for its anti-inflammatory properties and high content of monounsaturated fats, reduces plasma levels of IL-6 and CRP [10]. In the case of proteins, plant-based sources are associated with reduced circulating concentrations of CRP, TNF-α, and IL-6, while the consumption of red meats (beef, pork, lamb) correlates with elevated CRP and IL-6 levels [29]. This evidence underscores the significant influence of diet on inflammation and emphasizes the potential therapeutic implications of interventions aimed at modulating immune system activation. The following section provides a detailed description of the underlying mechanisms of isolated compounds, along with their primary clinical applications.

## 5. Effects of Vitamins and Minerals on Immune Function

Deficiencies in fat-soluble vitamins such as A, D, and E, as well as in water-soluble vitamins like folic acid, B6, B12, and C, impair immunity and reduce the body’s resistance to infections [33,34]. Addressing these deficiencies through targeted supplementation can effectively restore immune function to optimal levels. Likewise, several minerals, including zinc, iron, and selenium, have essential modulatory effects on immune components [34,35], and epidemiological studies indicate that their deficiencies are associated with increased morbidity [35,36]. However, excessive contents of both vitamins and minerals may adversely affect the immune response, underscoring the importance of careful monitoring, particularly with long-term supplementation [36]. This section examines the contributions of crucial vitamins and minerals to immune health, emphasizing their roles in cell regulation, mucosal integrity, and modulation of the inflammatory response, as well as the impact of deficiencies or excesses in disease susceptibility and progression.

### 5.1. Vitamin A

Vitamin A is crucial for maintaining immune system function, with effects mediated by the activation of gene expression through retinoic acid receptors and retinoid X receptors (RXR) [37]. These proteins are transcription factors that interact with specific DNA sequences, known as retinoic acid response elements, which regulate over 500 genes [38]. Vitamin A deficiency leads to the abnormal expression of structural proteins in epithelial tissues—such as the skin, respiratory tract, gastrointestinal tract, and genitourinary system—resulting in compromised barriers [39]. This condition is linked to a reduction in keratin, mucin, cilia, microvilli, and IgA, along with decreased cellularity in mucosal-associated lymphoid tissues (Figure 2) [40]. Inadequate levels of vitamin A impair hematopoiesis and affect the function of lymphocytes and myeloid cells. Generally, both deficient and excessive vitamin A predispose individuals to a Th1-skewed immune response and increase susceptibility to infections due to epithelial and mucosal dysfunction, particularly in the intestinal and respiratory tracts [41]. Supplementation with vitamin A in cases of deficiency has been shown to improve hematopoiesis, enhance mucosal integrity and function, and strengthen both the cellular and humoral immune responses. These effects, however, are generally not observed in well-nourished individuals [39,42]. In children infected with the human immunodeficiency virus (HIV) or malnourished individuals, vitamin A supplementation is associated with reductions in diarrhea, respiratory infections, and mortality rates [43,44,45]. Vitamin A also shows potential in the management of autoimmune and chronic inflammatory conditions. It has been shown to promote the induction of Tregs mediated by TGF-β, counteracting the effects of Th17 cells, a pro-inflammatory subset of CD4+ lymphocytes that produce IL-17, IL-21, and IL-22 [46]. Promising results have been observed in models of lupus, autoimmune encephalomyelitis, rheumatoid arthritis, and irritable bowel disease, although these findings require further validation in clinical settings [47].

### 5.2. Vitamin D

Historically, the primary diseases associated with low vitamin D levels have been skeletal disorders. Rickets in infants and osteomalacia in adults are well-established consequences of inadequate vitamin D stores [48]. The recognition that vitamin D also serves as a crucial stimulant of innate immunity emerged from reports on the treatment of tuberculosis with cod liver oil [49]. Low availability of calcitriol has been correlated with increased mortality from upper respiratory tract infections, including influenza, chronic obstructive pulmonary disease, and allergic asthma [50]. Furthermore, recent epidemiological studies also indicate that vitamin D deficiency is linked to the development of autoimmune diseases. For example, low serum calcitriol status has been shown to predict the onset of multiple sclerosis, type 1 diabetes, and rheumatoid arthritis. In addition, even when autoimmunity is established, vitamin D deficiency exacerbates its progression [51,52,53]. Vitamin D modulates the immune system in its active form, calcitriol, by interacting with vitamin D nuclear receptors (VDR) expressed on B and T lymphocytes, neutrophils, monocytes, and dendritic cells [54]. Calcitriol induces the production of antimicrobial peptides, such as defensins, and the antimicrobial peptide cathelicidin by macrophages, monocytes, keratinocytes, and epithelial cells in the intestines and lungs [55]. It enhances chemotaxis, phagocytic capabilities, and the antimicrobial functions of macrophages and monocytes (Figure 3). In epithelial cells, particularly in the intestines and lungs, calcitriol strengthens physical barrier function. Collectively, these antimicrobial effects support the body’s defenses against pathogens [55]. Conversely, vitamin D promotes a tolerogenic state in the adaptive immune system by transitioning from cell-mediated (Th1) immunity to humoral (Th2) immunity [56]. Calcitriol downregulates Th1 immune responses by inhibiting the production of pro-inflammatory cytokines such as IL-12, IFN-γ, IL-6, and TNF-α. In contrast, vitamin D enhances the secretion of type 2 anti-inflammatory cytokines, including IL-4, IL-5, and IL-10. This regulation is primarily mediated through the inhibition of NF-κB activation, facilitated by the enhanced expression of its inhibitory protein IkBα [57]. Additionally, it shifts the maturation of T cells away from the inflammatory Th17 phenotype and promotes the induction of Tregs. These effects are allowed by the inhibition of dendritic cell differentiation and maturation, preserving an immature phenotype, as evidenced by decreased expression of class II MHC receptors and costimulatory molecules [58]. This results in reduced production of the inflammatory cytokines IL-17 and IL-21, along with increased secretion of the anti-inflammatory cytokine IL-10, thereby promoting tolerance. Consequently, the overall effect of vitamin D is considered anti-inflammatory, making it beneficial in the prevention and management of autoimmune conditions [58]. Clinical trials investigating the use of vitamin D supplements have demonstrated favorable results on immune function, particularly in the contexts of autoimmunity, COVID-19, and chronic degenerative diseases [12,51,59].

### 5.3. Vitamin E

Vitamin E deficiency occurs in patients with intestinal malabsorption syndrome and negatively impacts both humoral and cell-mediated immune function [60]. Although studies are limited, they accentuate the essential role of vitamin E in immunity, highlighting impaired T-cell activity and reduced IL-2 levels [61,62]. Vitamin E is a potent lipid-soluble antioxidant, found in higher concentrations in leukocytes compared to other blood cells. It modulates immune cell function by directly influencing membrane integrity, signal transduction, and the production of inflammatory mediators by other leukocytes [63]. Furthermore, vitamin E regulates signaling pathways sensitive to oxidative stress, thereby mitigating NF-κB activation and the production of pro-inflammatory cytokines [63]. In this context, the reduction in free radicals due to vitamin E inhibits the enzymatic conversion of arachidonic acid to prostaglandins via COX-2 activity (Figure 3). Therefore, the production of PGE2 in macrophages decreases and enhances T-cell function and IL-2 secretion [64]. Vitamin E has been shown to positively influence humoral and cellular immune responses. For instance, lymphocyte exposure to vitamin E increases proliferation, immunoglobulin production, and IL-2 secretion [65]. Supplementation with vitamin E at doses greater than the dietary recommendations has been found to improve immune system function, particularly mediated by the activities of NK cells and neutrophils. Furthermore, its prophylactic consumption reduces the risk of infections and allergies, especially in older adults [66]. In addition, vitamin E supplementation enhances macrophage phagocytic capacity, thymic T-cell differentiation, and helper T-cell activity [67]. Recent studies have demonstrated that vitamin E supplementation reduces lipid peroxidation and pro-inflammatory markers in patients with moderate to severe COVID-19 pneumonia [68]. Furthermore, the incorporation of vitamin E was related to improved prognoses and shorter hospitalization durations. Numerous findings during the COVID-19 pandemic underscored the observation that the serum status of antioxidant micronutrients can decline during infection, highlighting the importance of adequate nutritional support for at-risk populations with metabolic vulnerabilities [12]. In summary, vitamin E deficiency impairs immune function, but supplementation above current dietary recommendations appears to enhance immunocompetence [62].

### 5.4. Vitamin C

Vitamin C deficiency leads to scurvy and increases susceptibility to potentially life-threatening infections [69]. Scurvy is characterized by the weakening of collagen structures, resulting in impaired wound healing and compromised immune function [70]. Infections also deplete vitamin C levels due to the increased metabolic demands generated by immune system activation. Combined with the body’s limited capacity to store water-soluble vitamins, this can result in lower plasma ascorbate content [70]. Vitamin C acts as a cofactor for lysyl and prolyl hydroxylases, enzymes essential for stabilizing the tertiary structure of collagen [71]. It also serves as a cofactor for two hydroxylases involved in the biosynthesis of carnitine, a molecule necessary for transporting fatty acids into mitochondria for energy production [72]. Additionally, vitamin C enhances epithelial barrier differentiation and function through the modulation of biosynthetic and signaling pathways, leading to increased lipid synthesis [73], the upregulation of tight junction proteins, and the prevention of cytoskeletal rearrangements [74]. As an antioxidant, vitamin C protects vital biomolecules from damage caused by oxidants produced during normal cellular metabolism and exposure to environmental pollutants due to its ability to donate electrons readily [75]. Notably, vitamin C neutralizes free radicals while also regenerating the antioxidants glutathione and vitamin E [76]. Vitamin C accumulates in phagocytic cells such as neutrophils and enhances chemotaxis, phagocytosis, the generation of reactive oxygen species, and, ultimately, microbial destruction [77]. Increased free radical generation activates the NF-κB pathway, leading to heightened inflammation and oxidative stress (Figure 3) [75]. Moreover, after phagocytosis or activation by soluble stimulants, vitamin C is depleted from neutrophils in an oxidant-dependent manner [78]. It is also essential for apoptosis and the removal of neutrophils from infection sites by macrophages, thereby reducing tissue damage [78]. The role of vitamin C in lymphocytes is less defined; it has been shown to improve B- and T-cell differentiation and proliferation, likely due to its effects on gene regulation [79]. Vitamin C supplementation is beneficial for the prevention and management of respiratory and systemic infections. Prophylactic use requires dietary intake that ensures adequate plasma concentration to optimize cellular and tissue availability [70]. In contrast, treating established infections requires significantly higher doses to compensate for the heightened inflammatory response and metabolic demand [80]. Since the prophylactic administration of vitamin C appears to reduce the risk of developing severe respiratory infections, such as pneumonia, it is likely that the low vitamin C levels observed during respiratory infections are both a cause and a consequence of the illness, resulting from increased inflammation and metabolic requirements [80].

### 5.5. Vitamin B

Folic acid, vitamin B12, and vitamin B6 are essential coenzymes that play a critical role in energy metabolism, protein synthesis, genetic material formation, and the epigenetic regulation of cells [81]. Consequently, deficiencies in these vitamins can compromise the renewal of epithelial linings and impair the proliferation and differentiation of immune cells. Specifically, deficiencies in folate and vitamin B12 increase susceptibility to infections by causing atrophy of lymphoid organs, including reduced cellularity in the thymus and bone marrow, as well as macroscopic alterations in epithelial tissues (Figure 2) [82]. These deficiencies lead to a reduction in circulating lymphocyte counts, decreased proliferative capacity of T cells in response to mitogens and antigens, and diminished amounts of IL-2 and antibodies. Furthermore, deficiencies in these vitamins reduce the phagocytic and bactericidal activity of polymorphonuclear leukocytes and suppress NK cell activity [83]. Vitamin B6 deficiency has been shown to suppress immune responses by affecting both humoral and cellular immunity. It induces atrophy of lymphoid organs and decreases the number and proliferative capacity of lymphocytes, as well as IL-2 production, while promoting a shift toward the Th2 phenotype, characterized by an increased amount of IL-4 [84]. Vitamin B12 plays a vital role in the conversion of methyl-tetrahydrofolate to tetrahydrofolate through a reaction catalyzed by the enzyme methionine synthase. A deficiency in vitamin B12 results in the accumulation of folic acid in its inactive form, methyl-tetrahydrofolate, which becomes trapped and cannot be reused, thus adversely affecting genetic material synthesis [82]. Both folate and vitamin B12 are required for the methylation of homocysteine to methionine in a reaction catalyzed by the enzyme methyl-tetrahydrofolate-homocysteine methyltransferase. Vitamin B12 and folic acid are fundamental to DNA methylation, with methyl-tetrahydrofolate donating its methyl group to vitamin B12 [85]. Subsequently, methylcobalamin acts as a cofactor for methionine synthase to facilitate the remethylation of homocysteine to methionine, which serves as a substrate for the production of S-adenosylmethionine, a compound essential for DNA methylation. This process is catalyzed by DNA methyltransferases that transfer methyl groups from S-adenosylmethionine to cytosine [85]. Furthermore, vitamin B12 ensures the structural stability of centromeres and subtelomeric DNA, both of which are critical regions of chromosomes [85]. Vitamin B6 also participates in the regeneration of methyl-tetrahydrofolate in the form of pyridoxal phosphate and is necessary for glycogen breakdown, acting in conjunction with glycogen phosphorylase. It is also involved in amino acid transformations as a coenzyme in transamination and decarboxylation reactions catalyzed by amino acid synthases or racemases [86]. Moreover, vitamin B6 influences tryptophan catabolism and the activity of the transcription factor NF-κB [87]. Recent research indicates that it acts as an anti-inflammatory agent by modulating inflammasome activity. These structures are a group of innate immune receptors that recognize patterns associated with pathogens or cellular damage, thereby activating inflammatory responses [88]. Low serum levels of vitamin B6 are commonly observed in patients with elevated inflammatory markers, and prior studies have demonstrated that supplementation with this vitamin produces anti-inflammatory effects and reduces oxidative stress [88]. The supplementation of folic acid and vitamin B12 enhances immune function and increases protection against infections, and, in cases of deficiency, reverses immunosuppression [83]. It is essential to recognize that these vitamins are water-soluble, making it crucial to prevent potential deficiencies. High-risk groups include patients with metabolic diseases associated with obesity, vegans, and individuals with chronic gastrointestinal disorders. Collectively, current research suggests that vitamin B6 could be utilized as an anti-inflammatory nutrient, while supplementation with vitamin B12 and folates should be carefully considered in conditions with an inflammatory basis unless a lower plasma status is identified [82].

### 5.6. Zinc

Zinc deficiency can result in growth retardation, impaired immune function, alterations in epithelial tissues, delayed wound healing, increased production of inflammatory cytokines, and heightened oxidative stress [34]. This essential element is critical for the structure and function of approximately 2800 macromolecules and over 300 enzymes [89]. Notably, the enzymatic activities of DNA polymerase, thymidine kinase, DNA-dependent RNA polymerase, terminal deoxynucleotidyl transferase, and aminoacyl-tRNA synthetase are zinc-dependent [89]. Furthermore, zinc ions help stabilize the functional structures of various proteins, including the family of transcription factors known as zinc finger DNA-binding proteins, the hormone thymulin, metalloproteinases, and superoxide dismutases [90]. In this context, the transcription factors metallothionein 1 and NF-κB possess zinc finger-like domains that are influenced by fluctuations in intracellular zinc concentration. Consequently, zinc status significantly impacts the proliferation, differentiation, and apoptosis of immune cells, as well as their microbicidal, antioxidant, and inflammatory activities, and their barrier functions [91]. Remarkably, zinc does not have a significant storage depot in the body, which means that deficiency can occur rapidly, with the immune system being particularly sensitive to changes in its levels [34]. Zinc deficiency is associated with atrophy of lymphoid organs and disrupts the structural integrity and function of epithelial membranes, including those in the skin and gastrointestinal tract, thereby increasing susceptibility to infection (Figure 2) [92]. A deficiency in zinc elevates the production of pro-inflammatory cytokines while decreasing the secretion of antibodies and antimicrobial peptides [92]. In polymorphonuclear cells, there is a reduction in chemotaxis and phagocytosis, inhibition of nicotinamide adenine dinucleotide phosphate oxidase activity, and impairment of extracellular trap formation. Consequently, the microbicidal processes involved in the elimination of phagocytosed pathogens and the capture of extracellular bacteria are hindered (Figure 3) [91]. Zinc deficiency leads to lymphopenia of T cells and negatively affects the maturation of lymphocytes, resulting in the accumulation of immature B cells and loss of lytic activity in NK cells. Additionally, such conditions may promote allergic responses due to polarization towards the Th2 phenotype [93]. Zinc supplementation has proven beneficial in managing viral infections caused by rhinoviruses, adenoviruses, coronaviruses, hepatitis viruses, and HIV [94,95]. It has also demonstrated positive effects against bacterial infections caused by *Shigella* and *Helicobacter*, parasitic infestations by *Leishmania* and *Plasmodium* [96,97], and autoimmune diseases such as type 1 diabetes mellitus [98]. While the prophylactic potential of zinc supplementation remains somewhat inconclusive, toxicity is unlikely; therefore, studies continue to establish the optimal consumption doses for a wide variety of conditions [34].

### 5.7. Iron

Iron is essential for numerous physiological and cellular processes; thus, its deficiency can lead to a variety of health consequences. The unique capacity of iron for oxidation and reduction makes it a critical mineral in many cellular redox reactions [35]. All cells require iron for the metabolism of amino acids, proteins, lipids, and carbohydrates, and for cellular proliferation. The common manifestations of iron deficiency include anemia, fatigue, reduced concentration, dizziness, pallor, and headache. Additional signs may comprise alopecia, dry skin, flattened nails, and glossitis [99]. The role of iron in immune function has been underscored by clinical observations linking its deficiency to increased susceptibility to infections. Research has demonstrated that specific immune responses are altered in the context of iron deficiency [100]. The mechanisms through which iron deficiency impacts immunity include a reduction in the activity of iron-dependent enzymes such as ribonucleotide reductase, which is essential for the biosynthesis of deoxyribonucleotide triphosphates required for DNA replication, and myeloperoxidase, which plays a crucial role in the bactericidal activity of neutrophils [101]. Furthermore, iron deficiency diminishes the gene expression of protein kinase C and phospholipase C, which are vital for the signaling transduction necessary for T lymphocyte activation. Similarly, it decreases the expression of cyclin A, a protein crucial for lymphocyte progression through the cell cycle, effectively arresting them in the G0/G1 phase [102]. Additionally, the production of reactive oxygen species, the activation of NF-κB, and cytokine secretion are all reduced [103]. Inadequate iron intake leads to atrophy of lymphoid organs (Figure 2) and increases overall susceptibility to infections by impairing innate immune responses mediated by T cells and antibodies [5]. The adverse effects of iron deficiency encompass diminished intracellular bactericidal activity of neutrophils due to decreased myeloperoxidase activity (Figure 3). Moreover, it reduces T lymphocyte counts, their proliferative response, and IL-2 production, resulting in impaired NK cell activity [5]. Ultimately, nearly every form of immune activation affects the mobilization and compartmentalization of iron in the body, generally leading to a reduction in its plasma content and storage within the mononuclear phagocytic system. This occurs because immune cells combat infections by sequestering iron [35]. Conversely, in the context of infections caused by intracellular pathogens, iron flows are reversed, shifting from phagolysosomes containing pathogens to the cytoplasm or extracellular environment [104]. The proteins involved in this process include ferroportin, ferritin, hepcidin, lactoferrin, and calprotectin. As long as immune stimuli and inflammatory mediators persist, iron restriction will lead to a functional shortage of mineral availability for hemoglobin synthesis in erythrocytes [35]. This, in turn, can lead to anemia due to chronic inflammation. It is important to note that while iron supplementation can address deficiencies, it must be carefully evaluated, as it can negatively influence the course of infection. There is considerable debate regarding the interaction between iron status, the administration of supplements, and susceptibility to infections. Although several studies suggest that preventing and treating iron deficiency can reduce the incidence of infections [105], iron overload can increase the risk of morbidity. Therefore, iron supplementation should be carefully monitored or avoided [106].

### 5.8. Selenium

Selenium is an essential micronutrient that plays a critical role in various aspects of human health, including thyroid hormone metabolism, cardiovascular health, and the prevention of neurodegenerative diseases and neoplasms, as well as ensuring an adequate immune response [107]. While severe selenium deficiencies are rare in humans, research conducted in animal models has indicated that such scarceness can lead to osteoarthritis, growth and reproductive issues, and cardiac and skeletal muscle myopathies [108]. Immune tissues such as the liver, spleen, and lymph nodes are rich in selenium [107,109]. The biological effects of selenium are primarily mediated through its incorporation into selenoproteins, which are crucial for the activation, proliferation, and differentiation of cells that drive both innate and adaptive immune responses [110]. Selenoproteins contain the amino acid selenocysteine, which is incorporated during protein synthesis. Their biosynthesis begins with the charging of serine onto a specific transfer RNA, which is subsequently phosphorylated to form selenocysteine through the donation of selenium from monoselenophosphate, facilitated by the enzyme selenophosphate synthetase [110]. The defining characteristic of selenoproteins is the presence of selenocysteine; however, the functional roles of this amino acid are diverse. Its biological functions include transcriptional regulation, phospholipid synthesis, protein folding, the reduction in methionine sulfoxide, and the biosynthesis of other selenoproteins [111]. Leukocytes express several, but not all, members of the selenoprotein family. The best-characterized selenoproteins relevant to immune function include glutathione peroxidases, thioredoxin reductases, iodothyronine deiodinases, methionine-R-sulfoxide reductase B1, and selenophosphate synthetase 2 [107]. For this reason, selenium is often described as an antioxidant, as it is a fundamental component of the enzymes responsible for detoxifying peroxides and peroxynitrites, and for donating electrons [109]. Selenium deficiency decreases the expression of selenoproteins, resulting in increased morbidity and mortality in response to various viral infections, potentially enhancing their virulence [112]. In murine models, selenium scarcity has been shown to increase genomic mutations in influenza and coxsackie viruses, leading to the emergence of more virulent strains and severe disease outcomes [113]. Lower serum selenium concentrations have also been linked to the accelerated progression of HIV and higher mortality rates among infected individuals [114]. Furthermore, selenium deficiency induces morphological changes in the bone marrow and thymus, characterized by decreased cellularity, as well as alterations in epithelial cells that compromise the barrier functions of mucous membranes, resulting in increased oxidative stress and inflammation (Figure 2) [115]. The lack of selenoproteins enhances oxidation, leading to increased lipid peroxidation and elevated superoxide anion levels in immune cells. This induces NF-κB activity and the enzymatic functions of COX and LOX proteins, culminating in the overproduction of pro-inflammatory cytokines and eicosanoids (Figure 3) [109]. Additionally, it impairs neutrophil function, leading to decreased total counts and diminished migratory, phagocytic, and microbicidal activities, the latter being associated with the abnormal enzymatic production of hydrogen peroxide. Similarly, T cells exhibit impaired proliferation in response to T-cell receptor stimulation due to their inability to suppress reactive oxygen species production when a person is deficient in selenoproteins [116]. The effects of selenium deficiency on immune function also include reduced serum amounts of IL-2, antibodies, and reduced total counts of T, B, and NK cells, along with diminished cellular activity [116]. Supplementation with selenium has been shown to exert immunostimulatory effects, such as enhancing the proliferation of activated T cells and increasing the activity of cytotoxic lymphocytes and NK cells. In elderly patients, it raises total lymphocyte counts, and it improves treatment responses in cancer patients [117]. Furthermore, selenium provides protective benefits against viral infections, enhances delayed-type hypersensitivity responses, and augments vaccine protection [118]. In addition, it promotes polarization toward the Th1 response, resulting in higher expression of IFN-γ, which is advantageous for immune responses against viral infections and tumors [117,119]. Finally, there is growing evidence supporting the potential benefits of selenium supplementation in inflammatory bowel disease, autoimmune disorders, asthma, and allergies [120,121].

## 6. Fatty Acids and Immunity

Fatty acids serve essential roles as components of biological membranes and as energy sources. Moreover, they are metabolized into bioactive molecules that modulate the functions of immune cells [15]. The essential fatty acids, linoleic acid (LA) and alpha-linolenic acid (ALA), cannot be synthesized by the human body. Deficiencies are common in patients with intestinal disorders or those on specialized feeding formulas. They may manifest as hyperproliferation of skin and mucosal cells, leading to symptoms such as dermatitis, eczema, erythema, coarse and sparse hair, frequent bowel movements, delayed wound healing, and impaired immune function [122,123]. The immunomodulatory effects of fatty acids depend on their chemical structure, as this influences their interactions with membranes, receptors, transcription factors, and enzymes [22]. Fatty acids modulate immune responses through multiple mechanisms, including interactions with G protein-coupled receptors (GPRs); Toll-like receptors (TLRs); cytosolic phospholipase A2, COX, and LOX enzymes; RXRs; peroxisome proliferator-activated receptor gamma heterodimer; and transcription factor NF-κB [123]. Generally, omega-3 polyunsaturated fatty acids (PUFAs) reduce oxidative stress and pro-inflammatory cytokine levels while enhancing anti-inflammatory mediators, including maresins, resolvins, and protectins derived from docosahexaenoic acid (DHA), and series-3 prostaglandins, series-5 leukotrienes, and resolvins derived from eicosapentaenoic acid (EPA) (Figure 3) [124,125]. Conversely, omega-6 fatty acids are primarily pro-inflammatory due to their metabolism into arachidonic acid, the precursor to pro-inflammatory mediators like LTB-4 and PGE2, via COX and LOX enzymes. Omega-3 fatty acids compete with arachidonic acid for these enzymes, thereby reducing the production of highly inflammatory eicosanoids. Additionally, DHA and EPA inhibit COX-2 activation and the subsequent release of prostaglandins [22]. Saturated fatty acids, such as lauric, palmitic, and stearic acids, induce inflammation by activating TLRs and NF-κB, which increases the secretion of pro-inflammatory cytokines like IL-1β, IL-18, and TNF-α [126]. Omega-3 fatty acids, especially ALA, DHA, and EPA, exhibit an inhibitory effect on the activation of both innate and adaptive immune cells; however, they support specific immune functions within certain cells. In macrophages and neutrophils, for instance, omega-3 fatty acids enhance phagocytosis, migration, and resolvin production while reducing cytokine secretion [127,128]. In contrast, they decrease the activation of basophils, dendritic cells, and lymphocytes, promote differentiation toward the Treg phenotype, and reduce the proliferation, activation, and differentiation of Th1 and Th17 cells (Figure 4) [129]. Saturated fatty acids lead to the activation and differentiation of T cells into pro-inflammatory phenotypes, whereas their impact on B cells remains unclear. Palmitic acid, for example, suppresses B-cell activation, but this effect is mitigated by EPA or DHA, which prevents the production of pro-inflammatory cytokines [126]. Supplementation with omega-3 PUFAs is regarded as a safe and effective approach to reducing inflammation. It has demonstrated benefits in managing various conditions, including coronary diseases, hypertension, type 2 diabetes, rheumatoid arthritis, lupus, inflammatory bowel diseases, and chronic obstructive pulmonary disease [124,130,131,132,133]. Omega-3 supplements, typically derived from fish oil, provide EPA and DHA directly. Additionally, ALA, which can be obtained from plant-based sources, is also beneficial as it can be elongated and desaturated into EPA and DHA in the human body, offering similar advantages for chronic disease prevention and management [22]. It is important to note that the Western diet generally contains high levels of saturated fats and pro-inflammatory omega-6 fatty acids, highlighting the necessity to modify dietary habits to optimize the health benefits associated with omega-3 fatty acid supplementation [134].

## 7. Immunomodulation by Phenolic Compounds

Plants synthesize phenolic compounds in response to various abiotic and biotic stressors, and these molecules are well regarded for their antioxidant properties and associated health benefits [135]. Recent research indicates that the immunomodulatory effects of phenolic compounds are mediated through their interactions with enzymes, transcription factors, and signaling pathways involved in responses activated by inflammation, detoxification, or oxidation [136]. Broadly, phytochemicals regulate eicosanoid synthesis by interacting with COX and LOX enzymes, and they suppress the transcription of pro-inflammatory genes by inhibiting NF-κB activation (Figure 4) [137]. Additionally, they influence multiple signaling pathways, including those mediated by phosphatidylinositol 3-kinase/protein kinase B, inhibitor of kappa B kinase/c-Jun N-terminal kinases, Janus kinase/signal transducer and activator of transcription, and mammalian target of rapamycin complex 1, which regulates cell proliferation and protein synthesis [138]. Their antioxidant properties are further linked to their capacity to activate the nuclear factor erythroid 2-related factor, which enhances cellular defense mechanisms [138]. Moreover, phenolic compounds serve as ligands for the aryl hydrocarbon receptor (AhR), a transcription factor that interacts with NF-κB and the xenobiotic response element, allowing them to modulate gene expression related to inflammation, drug metabolism, and toxin processing [139]. AhR activation by phenolic compounds also influences the differentiation of various immune cell types, including dendritic cells, macrophages, and NK, B, and T cells [139]. Collectively, phenolic compounds reduce the concentration of IL-6, TNF-α, CRP, PGE2, and reactive oxygen species while increasing the presence of tolerogenic dendritic cells, Tregs, and anti-inflammatory cytokines (Figure 4) [138]. Traditionally viewed as anti-nutrients, phenolic compounds are now increasingly recognized as essential micronutrients with significant health implications [140]. During infections, macrophages and pathogens stimulate the production of pro-inflammatory mediators, such as IL-6, IL-8, and reactive oxygen species, as well as prostaglandins. The overproduction of cytokines also contributes to the severity of inflammatory diseases, including asthma, atherosclerosis, diabetes, allergies, and cancer [138]. Therefore, managing inflammation—whether it arises from infection, obesity, dietary factors, chemical agents, or injury—is crucial, as it accelerates the onset and progression of degenerative diseases [140]. Furthermore, the immunomodulatory effects of polyphenols include antiviral and antimicrobial actions, highlighting their role in plant defense (Figure 4). Research has demonstrated that polyphenols can inhibit viral entry, replication, and release. In the case of bacteria, they disrupt cellular membrane integrity and permeability, interfere with intracellular enzymes involved in energy metabolism, and inhibit the synthesis of DNA, RNA, and proteins [136]. Notably, phenolic compounds not only mitigate the inflammatory processes underlying chronic degenerative diseases but also target specific proteins involved in conventional drug therapies [141]. For example, phenolics inhibit glycosidases, useful in diabetes management [142], and angiotensin-converting enzyme, relevant to hypertension and COVID-19 [143,144]. They also modulate cyclin-dependent kinases, COX enzymes, and NF-κB in cancer, and target AhR, offering therapeutic potential for autoimmune disorders, allergies, and food intolerances [139,145]. Prominent phenolic compounds include quercetin, epigallocatechin gallate, curcumin, and resveratrol, naturally abundant in foods like onions, green tea, turmeric, and grapes, respectively [136]. Although supplementation of these molecules has demonstrated therapeutic effects, evidence suggests that regular consumption of phenolic compounds from plant-based foods offers greater long-term benefits. This may be attributed to synergistic interactions among diverse bioactive phytochemicals present in whole foods [10].

## 8. Probiotics and Mucosa-Associated Lymphoid Tissue

The gastrointestinal microbiota comprises bacteria, archaea, fungi, algae, and protists. Recently, it has been suggested that this microbiome also includes phages, viruses, plasmids, and mobile genetic elements [146]. Research on the microbiome has progressed rapidly, leading to its recognition as an essential component of human health. The microbiota is established in early life, influenced by factors such as gestational age, delivery method, breastfeeding, weaning, and environmental factors [147]. The microbiota is generally stable in adulthood; however, it varies among individuals based on lifestyle factors such as diet, exercise, and body mass index. A balanced microbiota composition is critical for metabolism, immunity, and disease prevention [147]. In the gut, the most studied component is the bacterial microbiota, which includes major phyla such as *Actinobacteria*, *Proteobacteria*, *Fusobacteria*, *Verrucomicrobia*, *Firmicutes*, and *Bacteroidetes*. The latter two phyla together comprise approximately 90% of the gut microbiota. Within these, *Firmicutes* is highly diverse, encompassing the genera of *Lactobacillus*, *Bacillus*, *Enterococcus*, *Ruminococcus*, and *Clostridium*, while *Bacteroidetes* mainly includes *Bacteroides* and *Prevotella* [139]. Shifts in microbiota composition are essential for maintaining health across the lifespan, with evidence showing that diet strongly influences the microbiota. The Western diet, for instance, is associated with an imbalance or disruption in the normal composition and function of the microbiota, which, in turn, is linked to chronic degenerative diseases [148]. Probiotics, defined as live microorganisms beneficial to the host, are often part of the gut microbiota and support digestive, metabolic, and immune functions [149]. They promote immune health by modulating mucosa-associated lymphoid tissue and have been used therapeutically to regulate gastrointestinal mucosa and systemic immunity [149]. Probiotic interactions with host cells involve receptors, enzymes, and transcription factors responsive to microbial components or metabolites, such as TLRs, GPRs, AhRs, and histone deacetylases, which are crucial in gene regulation by removing acetyl groups from histones [139]. These pathways collectively increase the secretion of IgA, cytokines, and antibacterial agents, thereby strengthening gut barrier function [150]. Probiotics also support intestinal health by competing with pathogens for adherence to enterocytes [150]. Compounds within probiotic cell walls act as antigens that cross the intestinal epithelium to Peyer patches, influencing systemic immunity through innate and adaptive responses [151]. Their metabolic products further interact with enterocytes, dendritic cells, and lymphocytes, modulating immune cell signaling and cytokine and chemokine production, thus influencing the leukocyte subtypes involved in inflammatory, anti-inflammatory, allergic, or tolerogenic responses depending on the probiotic strain and diet [152]. The immunomodulatory and health benefits of probiotics have sparked interest in using them for disease treatment [153]. Probiotics enhance IgA levels in mucosal tissues and improve immune responses to vaccination [154]. They have demonstrated positive effects on NK cell activity and are beneficial in managing chronic and antibiotic-associated diarrhea [155]. Clinical evidence supports the efficacy of probiotics in conditions such as irritable bowel syndrome, ulcerative colitis, Crohn’s disease, allergies, and dermatitis [155,156,157]. The effects of probiotics can vary by species and strain, with *Lactobacillus*, *Bifidobacterium*, and *Streptococcus* showing significant clinical benefits [153]. Probiotics with immunostimulatory properties combat infections and cancer cells by inducing IL-12, which activates NK cells and promotes Th1 cell development. They also regulate allergic responses by balancing Th1 and Th2 cells [150]. Conversely, immunoregulatory probiotics promote IL-10 and Treg production (Figure 4), which mitigates allergies, irritable bowel syndrome, and inflammatory responses [158,159]. Probiotic therapy holds promise in modulating the gut microbiota and immune responses. However, while numerous studies highlight their health benefits, ongoing research is necessary to clarify the effects of specific probiotics on immune parameters and their potential role in disease prevention and management [17].

## 9. Conclusions

Nutritional status and the intake of specific nutrients have significant effects on the tissues, cells, components, and functions of the immune system. Consequently, targeted nutrition and supplementation offer potential therapeutic benefits in managing infectious, metabolic, and chronic degenerative diseases by modulating immune responses and influencing disease progression. The COVID-19 pandemic underscored the critical role of immunonutrition in patient care, as deficiencies in essential nutrients and obesity were associated with increased risks of acute respiratory distress syndrome and mortality. Evidence suggests that supplementation with vitamin D, vitamin C, and omega-3 fatty acids can improve patient outcomes by reducing serum pro-inflammatory cytokines, shortening hospital stays, decreasing the need for respiratory support, and lowering the risk of severe COVID-19 and intensive care unit admissions. Despite these promising findings, further research is needed to establish optimal dosages and standardize therapeutic protocols across various conditions. Therefore, immunonutrition is an emerging field that holds significant potential to become a crucial area of interest for healthcare professionals seeking to promote immune health and improve patient care outcomes.

## Figures and Tables

**Figure 1 nutrients-16-04363-f001:**
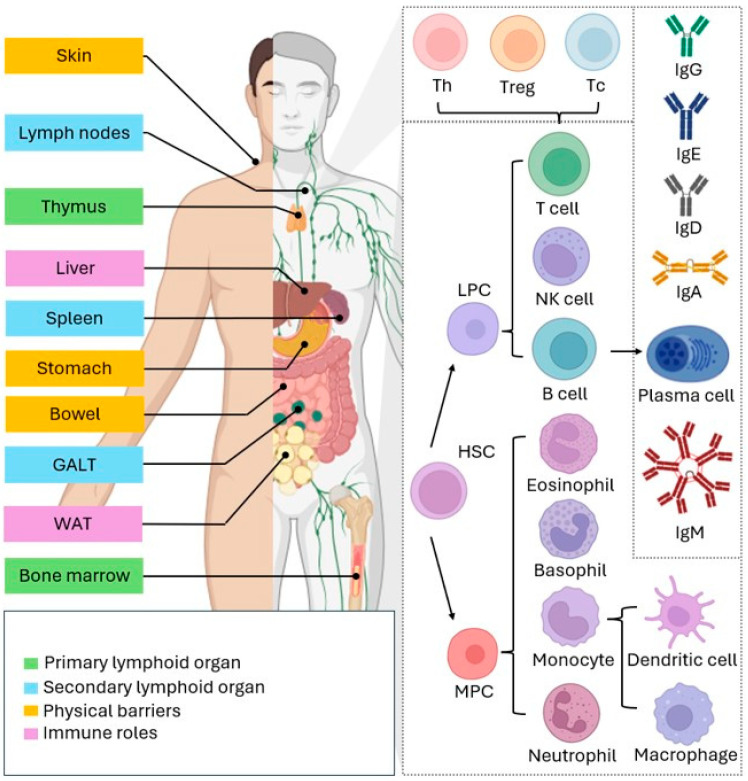
A schematic representation of the anatomical localization of the components of the immune system and leucocyte differentiation in the bone marrow and thymus. The skin and mucosal membranes form the first line of defense, acting as physical barriers that prevent pathogen entry. These surfaces are coated with beneficial microorganisms that secrete antimicrobial substances and mucus to inhibit pathogen colonization. Immune cells originate from hematopoietic stem cells (HSCs) in bone marrow, differentiating into myeloid progenitor cells (MPCs) or lymphoid progenitor cells (LPCs) based on microenvironmental signals. MPCs produce eosinophils, basophils, neutrophils, and monocytes, with monocytes further maturing into dendritic cells or macrophages. LPCs give rise to T cells, B cells, and NK cells. T cells complete their differentiation in the thymus, becoming regulatory (Treg), helper (Th), or cytotoxic cells (Tc), while B cells mature in the bone marrow and later differentiate into plasma cells that secrete antibodies upon antigen exposure. Secondary lymphoid organs support antigen presentation, lymphocyte activation, and immune response generation. Gut-associated lymphoid tissue (GALT) is the largest component of the immune system, while the liver and white adipose tissue (WAT) contribute to immunity by producing innate immune proteins, cytokines, and immunomodulatory molecules.

**Figure 2 nutrients-16-04363-f002:**
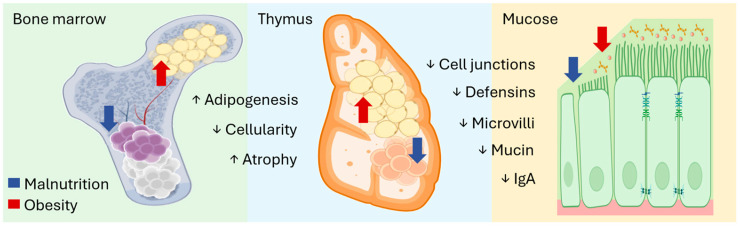
Malnutrition and obesity profoundly affect the bone marrow, thymus, and mucosa, disrupting their structure, cellular composition, and function. In primary lymphoid organs, malnutrition induces atrophy and hypocellularity, significantly reducing hematopoietic stem cells and thymocytes. In the mucosa, it leads to epithelial atrophy and impaired integrity, which compromises secretion, absorption, and cellular renewal. Obesity, on the other hand, increases adiposity in the bone marrow and thymus, accelerating age-related involution and impairing lymphopoiesis and immune function. It also heightens mucosal inflammation and permeability. Deficiencies in zinc, iron, selenium, and B vitamins lead to the bone marrow and thymus changes seen in malnutrition, while deficits in vitamins A, D, and C are linked to mucosal dysfunction. Importantly, immune organ impairments associated with obesity are induced and further exacerbated by high-fat or Western diets (detailed information is provided in the following sections).

**Figure 3 nutrients-16-04363-f003:**
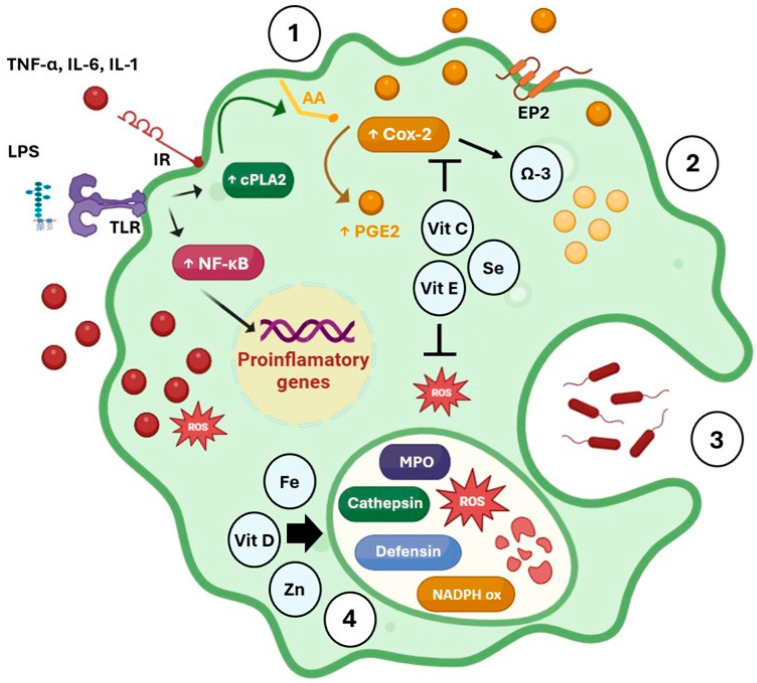
Macrophage activation (1) involves migration (2), phagocytosis (3), and the destruction of engulfed antigens (4). Pro-inflammatory signals such as cytokines or lipopolysaccharides (LPS) activate Toll-like (TLR) or interleukin receptors (IR), respectively, leading to the translocation of NF-κB to the nucleus, enhancing the transcription of pro-inflammatory genes. Simultaneously, cytosolic phospholipase A2 (cPLA2) releases arachidonic acid (AA) from membrane phospholipids, which is metabolized by cyclooxygenase-2 (COX-2) into prostaglandin E2 (PGE2). PGE2 acts in a paracrine manner through prostaglandin E2 receptor (EP2), further activating NF-κB and amplifying inflammatory responses. The cytokines produced reinforce pro-inflammatory activity by increasing oxidative stress and recruiting additional immune cells, thereby amplifying inflammation. Nutrients play a critical role in modulating these pathways. Vitamin C, vitamin D, and selenium exert antioxidant effects, inhibiting COX-2 activity and reducing oxidative damage caused by the respiratory burst that liberates reactive oxygen species (ROS). Omega-3 fatty acids (Ω-3) compete with AA for COX-2, producing PGE3, resolvins (D-series and E-series) that promote inflammation resolution. Vitamin D enhances the expression of antimicrobial peptides such as defensins and cathepsins. Iron supports the activity of myeloperoxidase (MPO), whereas zinc is vital for the functionality of the NADPH oxidase complex, both enzymes crucial to producing hypochlorous acid and superoxide to kill pathogens in the phagolysosome. Immune components rely on these nutrients for optimal function, and deficiencies impair the immune response, increasing susceptibility to infections or exacerbating collateral tissue damage caused by excessive inflammation.

**Figure 4 nutrients-16-04363-f004:**
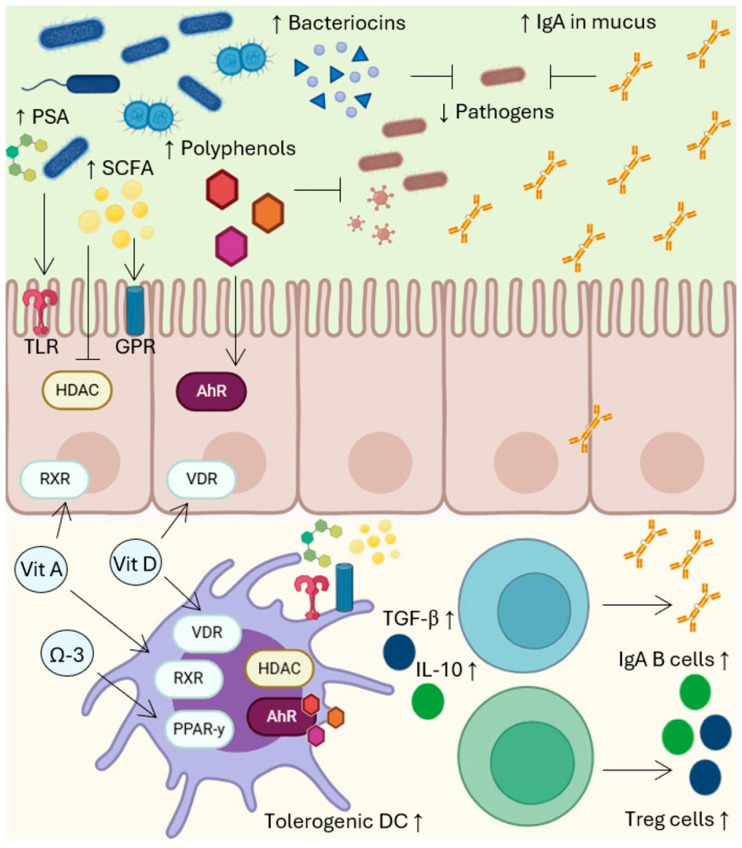
Commensal microorganisms and probiotics enhance mucosal integrity and promote anti-inflammatory processes in gut-associated lymphoid tissue. They are detected by receptors such as TLR (Toll-like receptor) and GPR (G protein-coupled receptor), enzymes like HDAC (histone deacetylase), and transcription factors such as AhR (aryl hydrocarbon receptor). Upon activation, these pathways stimulate epithelial differentiation and the production of structural proteins, defensins, and mucus, strengthening the mucosal barrier. Critical microbial products include polysaccharide A (PSA), short-chain fatty acids (SCFAs), phenolic compound metabolites, and bacteriocins. Unmetabolized polyphenols also exhibit antiviral and antimicrobial properties and promote epithelial differentiation. Probiotics and polyphenols further modulate immunity by driving the development of tolerogenic dendritic cells (DCs), which secrete anti-inflammatory cytokines like TGF-β and IL-10. DCs promote B-cell differentiation into IgA-secreting cells and induce T cells to adopt a regulatory phenotype (Tregs), creating an anti-inflammatory environment and enhancing barrier defenses. Additionally, omega-3 fatty acids (Ω-3), vitamin D, and vitamin A support tolerogenic DC development through their interactions with transcription factors peroxisome proliferator-activated receptor gamma (PPAR-γ), RXR (retinoic X receptor), and VDR (vitamin D receptor), respectively. These vitamins are also crucial for mucosal differentiation and function, emphasizing their essential role in maintaining gastrointestinal health.

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
