# Peer review of "Dietary Modulation of the Immune System"

_nutrients, 2024, doi:10.3390/nu16244363_

Round 1
Reviewer 1 Report
Comments and Suggestions for Authors
This review summarizes the effects of nutrition on immune system activity specifically the role of micronutrients, fatty acids, and phenolic compounds in promoting proper immune system function. The review was well written and easy to read.
The authors discussed the importance of various dietary components for immune system function and the effects of nutrient deficiencies or excesses.
The topic, though not original, effectively summarizes the impact of various dietary components on the immune system, providing valuable information that can inform and inspire the scientific community.
This topic may benefit the scientific community, as nutrient alterations could be linked to disease development and progression and may serve as therapeutic interventions to enhance current efficacy strategies.
I recommended the authors include figures to visually summarize the text.
This review highlights the critical role of proper nutrition in maintaining an effective immune system, which is vital for disease prevention and improved prognosis.
Author Response
Dear Reviewers,
Thank you for providing valuable feedback and the opportunity to improve our manuscript. We have carefully addressed your comments and recommendations, incorporating significant revisions to enhance the clarity and impact of our work. Notably, we have included four figures to visually illustrate key concepts that were previously explained only in text. Below is a summary of the changes:
Figure 1: This figure depicts the anatomical localization of immune system components, including primary lymphoid organs, secondary lymphoid organs, physical barriers, and the immune roles of other tissues. Additionally, it illustrates the differentiation of hematopoietic stem cells in the bone marrow into major leukocyte types, the subdifferentiation of T cells in the thymus, and the immunoglobulin classes produced by plasma cells.
Figure 2: This figure highlights the key effects of malnutrition and obesity on organ size, cellularity, structural integrity, and secretions. It also introduces the shared nutrient deficiencies that can induce similar changes in these tissues, as further detailed in subsequent sections.
Figure 3: This visual summary outlines the roles of vitamins, minerals, and omega-3 fatty acids during macrophage activation and phagocytosis. It emphasizes their critical functions as antioxidants, microbicidal agents, and mediators of inflammatory resolution.
Figure 4: This figure illustrates the intricate interactions between commensal microorganisms, probiotics, phenolic compounds, and epithelial integrity. It also details their immunomodulatory roles in gut-associated lymphoid tissue (GALT), supporting both epithelial and immune functions.
We have taken great care to ensure that the illustrations are accurate and faithful, reflecting the anatomical, histological, cellular, and molecular processes they depict. In addition to these substantial changes, we identified and corrected minor grammatical errors, including punctuation, spacing, and formatting inconsistencies, to improve the overall quality and readability of the manuscript.
We believe these revisions have significantly enhanced the manuscript's quality and its contribution to the field. We kindly request your consideration for acceptance of this revised version.
Kind regards,
Dr. Luis Fernando Méndez López
Universidad Autónoma de Nuevo León, Facultad de Salud Pública y Nutrición. Avenida Dr. Eduardo Aguirre Pequeño y Yuriria, Col Mitras Centro, Monterrey, Nuevo León, México, C.P. 66460. e-mail: luis.mendezlop@uanl.edu.mx

Reviewer 2 Report
Comments and Suggestions for Authors
In this review, Lopez et al. give an extensive and well-written overview over dietary, nutrients and malnutrition and their complex interplay with the immune system. In general, the article is very well-written and understandable. The only and most critical point is the complete absence of any figures, which are essential for the understanding of the content, especially with this massive body of text and information. Therefore, the authors have to provide at least three figures for the most central parts of the review, i.e. Section 2, 3, and 5. This must be addressed in a major revision and then this manuscript will be a helpful contribution to the field.
Author Response

(The authors gave the same response as above.)

Round 2
Reviewer 2 Report
Comments and Suggestions for Authors
The authors did a nice job. The figures are sufficient to support the already well-written manuscript. Thank you very much.